# The Impact of the Digital Capability of College Students' New Enterprises on Business Model Innovation Driven by the Digital Economy: The Mediating Effect of Digital Opportunity Discovery

**Fengliang Li * and Khunanan Sukpasjaroen ***

Chakrabongse Bhuvanarth International Institute for Interdisciplinary Studies (CBIS), Rajamangala University of Technology Tawan-ok, Bangkok 10400, Thailand

\* Correspondence: fengliang.li@rmutto.ac.th (F.L.); hhkhuna@gmail.com (K.S.)

**Abstract:** Based on the theoretical frameworks on dynamic capabilities and business model innovation, we conducted a comprehensive survey and analysis involving 451 Chinese university student enterprises. The primary objective was to investigate the synergistic mechanism between these two factors, assessing their impact on business model innovation and tracing the evolutionary path. The study revealed the following key findings: (1) positive correlations exist between digital capabilities and business model innovation; (2) entrepreneurial passion serves as a mediator in the positive relationship between digital capabilities and the discovery of digital opportunities; (3) digital opportunity discovery acts as a mediator in the relationship between digital capabilities and business model innovation; (4) under the mediation of dynamic capabilities, digital opportunity discovery significantly promotes business model innovation. Our research contributes to the empirical exploration of digitization in enterprises, shedding light on the collaborative influence of digital capabilities and digital opportunity discovery on business model innovation. Importantly, it elucidates the contextual boundaries that influence business model innovation through diverse pathways, enhancing our comprehensive understanding of the dynamic landscape in the evolution of digital business transformations.

**Keywords:** digital capability; business model innovation; new enterprises; digital economy

## 1. Introduction

The ongoing advancement of the digital economy has accelerated the transformation of both new and traditional forms of kinetic energy in economic development (X. Zhang 2019). Moreover, substantial shifts have transpired in the allocation and organization of corporate innovation resources (Wang et al. 2020). Business growth has experienced rapid acceleration due to innovations in business models (Wu et al. 2021). While previous research has proposed leveraging digital capabilities to foster business model innovation in new ventures, the existing perspective is predominantly theoretical and conceptual, marked by fragmented content and a dearth of in-depth exploration and empirical research on the correlation between digital capabilities and business model innovation. Consequently, it is imperative to conduct a comprehensive analysis of the mechanism through which digital capabilities influence business model innovation.

Digital opportunity discovery plays a pivotal role in establishing a connection between digital capability and business model innovation. Nevertheless, the literature needs to address and explore the crucial pathway associated with digital opportunity discovery in relation to digital capability and business model innovation. By elucidating how digital capability leads to business model innovation, we can provide further clarity on the process-based relationship between digital capability and business model innovation (Dong 2021).

In summary, digital capability, as the focal point of entrepreneurship research in the digital age, holds theoretical and practical significance for new ventures to actively discover and create digital opportunities, thereby propelling the innovation of new business models. However, a comprehensive understanding of the influence mechanism of digital capabilities on startup business model innovation can only be achieved through a synthesis of existing practices and theoretical developments. Therefore, the research questions addressed in this paper are as follows:

Firstly, in the digital environment, does the digital capability of new ventures significantly impact business model innovation? What specific action path defines this influence? Secondly, drawing from dynamic capability theory and innovation theory, can we incorporate digital opportunity discovery, the entrepreneurial passion for invention, and the dynamic capability of enterprises as influencing factors in the relationship between digital capability and business model innovation, thus revealing the influence mechanism of digital capability on business model innovation? Finally, what functional relationships exist among the variables involved? Analyzing these issues is crucial for illustrating the impact of digital ability on business model innovation, and it is essential to delineate the boundaries of business model innovation promoted by university student startups through digital capabilities.

This study formulates an empirical model that explores the impact of digital capabilities and digital opportunity discovery on business model innovation in college students' new ventures, combining the existing literature and empirical research. It aims to verify and enhance the research framework by addressing supplementary, moderating, and refining research questions and drawing corresponding conclusions. In this paper, I investigate the direct impact of digital capabilities on business model innovation from strategic and dynamic perspectives, along with the mediating role of digital opportunity discovery in the relationship between digital capabilities and business model innovation. Additionally, I analyze the moderating effects of the entrepreneurial passion for innovation on the relationship between digital capability and digital opportunity discovery, as well as the moderating effect of the dynamic capability of an enterprise on the relationship between digital opportunity discovery and business model innovation.

## 2. Materials and Methods

### 2.1. Theoretical Foundation and Research Hypotheses

2.1.1. The Importance of Digital Capabilities for Business Model Innovation

Experts have examined the concept of business models and digital capabilities from multiple research viewpoints. According to X. Zhang (2019), a business model is a value-creating and value-capturing activity system designed by an enterprise. Presently, domestic scholars primarily define business models from value creation and acquisition perspectives. Yang et al. (2020) defined it as the infrastructure for enterprises to create, transmit, and acquire value. Furthermore, scholars have largely agreed on the fundamental components of a business model, which encompass the value proposition and market segmentation, the value chain structure, the value acquisition mechanism, and the interrelationships between these elements (Saebi et al. 2016).

Digital capability was defined by Khin and Ho (2019) as a distinctive dynamic capability of new businesses working in a digital world. Only companies with strong digital capabilities are more likely to adopt digital technology and have the means to invest in developing it into new digital goods. Here, we define digital capability as an organization's capacity to handle expert knowledge and make use of digital technology, while creating new digital products. According to Zhu et al. (2020), digital ability is the bedrock of digital entrepreneurship and the critical capability determining how businesses develop digital opportunities. Digital technology and digital ability interact to provide digital opportunities. Digital capabilities have been shown to support and enhance businesses' digital entrepreneurial activities, according Khin and Ho (2019). Internet enterprises should have

digital functions conducive to promoting the transformation of internet enterprises into an organization's overall competitive advantage.

As new ventures often face challenges in expediting an enterprise's innovation process, dynamic capability theory posits that specific dynamic capabilities are instrumental in helping enterprises achieve their innovation goals within a constantly evolving entrepreneurial environment. The success of an enterprise's digital product development and business model innovation hinges strongly on its digital skills in today's digital, entrepreneurial landscape. Digital capability encompasses an entrepreneur's or new venture's comprehensive ability to adapt to developing new products or environmental changes in the digital sphere (Khin and Ho 2019). Enterprises possessing digital competence are more likely to leverage digital technology for new product development, enhanced customer experiences, and streamlined operational procedures, thereby driving, and realizing business model innovation (Westerman et al. 2012; Khin and Ho 2019). Digital capabilities can significantly benefit businesses by augmenting their strategic sensitivity and adaptability, thereby fostering innovation. From a strategic perspective, the strategic elements of digital capability empower new ventures to strategically manage their digital development (Zhu et al. 2020) and prompt businesses to innovate their business models toward digitization.

Entrepreneurs' capacity to reconstruct enterprise resources and innovate business models can be heightened through digital capability. Augier and Teece (2009) argued that businesses stand to gain from specific dynamic capabilities when faced with a dynamic environment. From a dynamic perspective, innovation theory asserts that an enterprise's resources and how they are reconstructed determine the enterprise's ability to innovate its business model during development (Winterhalter et al. 2015). The dynamic capability of digitization can aid new enterprises in dynamically allocating resources and enhancing the process of resource reconstruction in a digital environment (Autio et al. 2018).

In sum, I put forth the following assumptions:

**H1.** *Digital capability positively influences business model innovation.*

2.1.2. Intermediary Role of Digital Opportunity Discovery between Digital Capability and Business Model Innovation

According to Davidson and Vaast (2010), digital opportunities are a new type of entrepreneurial opportunity made possible by digital technology. These digital opportunities have altered the way entrepreneurial opportunities are generated, and many digital opportunities have been created as a result of information technologies, including mobile internet, artificial intelligence, and cloud computing. Identifying digital opportunities within entrepreneurial opportunity is the central and primary focus of entrepreneurial research, and opportunities are constantly being discovered, created, absorbed, conveyed, and aggregated as the central energy of the digital entrepreneurial ecosystem (H. Zhang 2018). The core subject drives other subjects in the digital entrepreneurial ecosystem to identify and exploit opportunities through information exchange, resource integration, and knowledge spillover, forming a complex collection of entrepreneurial opportunities.

The capability to identify corporate business model innovation through digital opportunities is, to a certain extent, linked to the success of a new venture's innovation and entrepreneurial activities (Yu et al. 2018). With digital capabilities, new ventures can focus their efforts on applying and developing digital technology, better identifying digital opportunities in the marketplace (Li et al. 2017), and capitalizing on the intrinsic value of digital opportunities, thereby promoting business model innovation.

Digital capability can aid entrepreneurs in enhancing their information perception, thus expediting the process of digital opportunity discovery. The term 'digital capability' denotes a business owner's ability to connect with the outside world through information channels, such as a digital platform or a digital ecosystem, expanding the market's information boundary and sources. This, in turn, enables entrepreneurs to discover digital opportunities in the market. Through the influence of digital ability on entrepreneurs'

active perceptions of their environment and information, entrepreneurs with solid digital abilities are more likely to leverage emerging information and communication technologies to perceive and discover digital opportunities more effectively.

Digital opportunity discovery can enhance entrepreneurs' innovation abilities, fostering enterprise business model innovation. According to Wang and Zhang (2022), business model innovation is the process through which enterprises identify and exploit opportunities for value creation. Enterprises innovate business models by recognizing and exploiting opportunities. Entrepreneurial enterprises, facing resource constraints and cost pressures, often engage in low-cost opportunity discovery activities, such as improving existing technologies and processes to boost innovation of products and business models (Guo and Shen 2014). The development of digital products and the exploration of digital technology are increasingly critical factors in the success of new ventures in an uncertain environment. Through digital opportunity discovery, new firms can acquire new knowledge, cultivate new products, enhance their innovation capacity, achieve value creation, and promote business model innovation.

Digital opportunity discovery, facilitated by digital capability, can drive business model innovation. According to dynamic capability and entrepreneurial opportunity theory, the dynamic capability of an enterprise is advantageous in identifying opportunities and accessing resources. Entrepreneurs can benefit from digital capability by establishing and developing their digital positioning, expanding their information sources, and more effectively identifying and evaluating digital opportunities. Identifying digital opportunities will guide enterprises in aligning their internal resources with opportunities or reconstructing their resources based on the discovered digital opportunities to maximize the opportunity value, thereby promoting business model innovation (Peng et al. 2019).

In sum, I put forth the following assumptions:

**H2.** *Digital capability positively influences digital opportunity discovery.*

**H3.** *Digital opportunity discovery positively influences business model innovation.*

**H4.** *Digital opportunity discovery plays an intermediary role between digital capabilities and business model innovation.*

### 2.1.3. Regulatory Function of an Entrepreneur's Passion for Invention

The interpretation by Ksenia et al. (2021) focuses on the definition on the entrepreneur's passion for invention and drive for innovation and invention. In the business world, a passion for invention can drive entrepreneurs to constantly try new ideas, products, or services to meet market needs or create new market opportunities. Kapoor and Kaura (2018) asserted that due to the characteristics of the digital economy ushered in by the mobile internet, firms must possess digitization and digital resource coordination capabilities to effectively utilize digital information and knowledge as a new element of production. Warner and Wäger (2019) introduced a dynamic capability model for digital transformation, delineating three capabilities, digital perception, digital acquisition, and digital transformation, all exemplifying digital transformation. Consequently, digital capability represents the static manifestation of an enterprise's distinct competence.

While numerous startups exhibit robust digital capabilities, in today's highly dynamic digital environment those with strong digital capabilities can direct their efforts towards the utilization and development of digital technologies, along with identifying digital opportunities in the market (Li et al. 2017). Nevertheless, it proves challenging to pinpoint digital opportunities in an unpredictable market solely relying on digital capabilities. Consequently, our aim is to introduce entrepreneurial passion for invention to modify and fortify the relationship between digital capability and digital opportunity discovery.

Entrepreneurial passion is a concept that pertains to the individual level of the entrepreneur. Inventive passion denotes the enthusiasm of entrepreneurs for identifying new

market opportunities, developing new products or services, experimenting with new methods, and engaging in other entrepreneurial activities (Zhou et al. 2021). Entrepreneurs with a passion for invention relish introducing new product or service concepts to the market. According to identity theory, individuals perceive themselves as a collection of identities based on specific roles and act in accordance with their assigned roles and identities; identity can thus shape, maintain, and guide individual behavior within a social system.

To commence, the entrepreneur's role as an inventor is inherently tied to their inventive passions. Entrepreneurs adopting the inventor role will translate their digital capabilities into digital opportunities, such as developing new products and launching new services to better cater to the external market's needs. This necessitates the design and modification of the company's management model and the framework connecting these elements, thereby aiding in motivating new enterprises to innovate their business models. For instance, Alibaba Group, an internet behemoth that prioritizes innovation, identified the rural e-commerce market as a target, formulated rural e-commerce strategies, and generated new entrepreneurial opportunities through its robust digital capabilities.

Concurrently, a passion for invention will significantly enhance entrepreneurs' abilities to creatively solve problems (Cardon et al. 2013), encompassing the creativity of entrepreneurs as inventors (Baron et al. 2016). This boost in creativity will assist enterprises in designing and altering the critical elements of business models and the framework connecting these elements, essentially aiding enterprises in advancing their business. For instance, the success of the Didi taxi company stems from creatively appropriating road information through digital capabilities, impacting product design and customer perceptions.

As a result of the foregoing analysis, digital capability can aid businesses in identifying digital opportunities by influencing the processes of environmental awareness and resource allocation creativity. While digital capability significantly fosters the relationship between enterprises and entrepreneurs, this progression demands that entrepreneurs maintain optimistic and proactive attitudes and emotions. Entrepreneurial passion has been demonstrated to significantly influence entrepreneurial behavior, with entrepreneurial passion more likely to uncover opportunities than other types of entrepreneurs (Baron 2008). Hence, entrepreneurial passion can amplify the impact of digital ability on the identification of digital prospects.

In sum, I put forth the following assumption:

**H5.** *Entrepreneurs' passion for invention positively regulates the relationship between digital ability and digital opportunity discovery.*

2.1.4. Role of the Dynamic Capabilities of Enterprises on Digital Opportunity Discovery and Business Model Innovation

Dynamic capabilities, according to Wang and Ahmed (2007), are a type of high-level capability, and emphasize enterprises' continuous pursuit of strategic iterations, including three dimensions: absorptive, adaptive, and innovative. The interaction between the environment and resource arrangement has a significant effect on how digitally mature businesses are (Chen and Tian 2022).

Zott and Amit (2007) proposed an efficient and innovative business model, asserting that choices can complement, rely on, or even transform one another. Efficiency-based business model innovation aims to reduce transaction costs for all participants, simplify transaction procedures, enhance organizational efficiency, boost participant reliance on core enterprises, and facilitate innovative activities (M. Qiu et al. 2021). Dynamic capability comprises two components: 'dynamic', referring to the constant change of the external environment, and 'capacity', denoting an enterprise's ability to integrate and reconstruct resources (He et al. 2017; Yao et al. 2022). Enterprises can influence business model innovation through digital opportunity discovery. On one hand, the process of discovering digital opportunities is a dynamic behavioral process (Y. Qiu et al. 2021), essentially involving information and knowledge discovery and acquisition. Enterprise dynamics positively impact knowledge acquisition, integration, and the development of new business models

(J. Zhang 2021). According to the summary on digital opportunity discovery in this paper, the digital opportunity for startups operating in a digital environment is also an innovation and entrepreneurship opportunity. According to the 'creation view', entrepreneurial opportunities are not objective but rather the result of entrepreneurs attempting to disrupt the market equilibrium, arising from an individual's subjective perceptions of the external market environment. New entrepreneurial opportunities are identified or created through creative actions (He et al. 2017; Guo and Hou 2021).

According to resource dependence theory, a complex and changing market environment will exert varying effects on enterprises' knowledge management activities, subsequently influencing business model innovation. The highly dynamic environment increases the difficulty for businesses to accurately identify, search for, and acquire necessary resources. In this context, acquiring external knowledge becomes crucial for coping with the environment. The business model innovation cycle is thereby shortened, and efficient knowledge acquisition expedites the pace of model innovation in business, ensuring process effectiveness (He et al. 2017). If businesses can integrate knowledge in real-time, master new information and technologies, and identify and exploit digital opportunities for adaptation, they will be better equipped to adjust to their environment. The role of enterprise dynamic capability adjustment in fostering business model innovation will become more apparent. Thus, resource dependence theory elucidates how an enterprise's dynamic capabilities play a moderating role in recognizing new digital opportunities and constructing new business models. Consequently, I propose the following hypothesis:

**H6.** *The dynamic capability of an enterprise positively regulates the relationship between digital opportunity discovery and business model innovation.*

According to the research hypotheses, I selected digital capability as the independent variable, business model innovation as the dependent variable, digital opportunity discovery as the intermediary variable, and the entrepreneur's passion for invention and the enterprise's dynamic capability as moderating variables. These five variables are expected to have a positive influence on business model innovation. Figure 1 illustrates this framework.

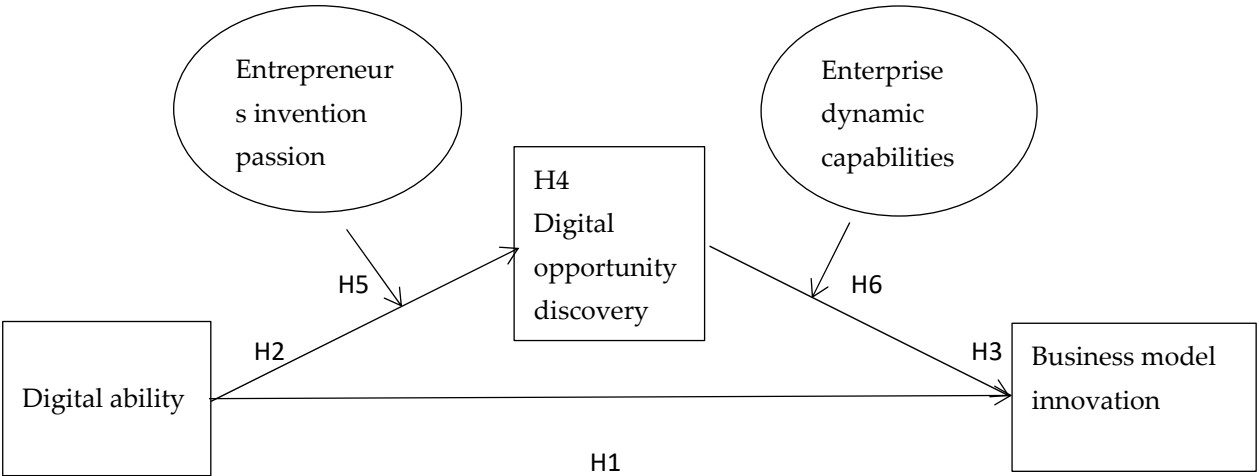

**Figure 1.** Empirical model of this study.

## 3. Methodology

### 3.1. Research Design

3.1.1. Sample and Data

According to the China Innovation and Entrepreneurship Index ranking compiled by the China Academy of Economic Research and the Key Laboratory of Big Data Innovation and Knowledge Management at the University of Chinese Academy of Sciences, along with

representative cities in different regions (northeast, north China, South China, northwest, southwest), enterprises were selected in five cities: Shenyang, Beijing, Shenzhen, Xi'an, and Chongqing. The selection was based on research data obtained from renowned creative spaces and incubation platforms, such as the *National Pioneer Park* and urban incubators in these cities (Wu et al. 2021). These cities and regions exhibit the best economic development. The companies chosen meet the research criteria of various reputable maker spaces, ensuring the quality of the research and its applicability. I specifically targeted startup companies with a five-year operating history (Li 2016). A total of 605 questionnaires were distributed to the executives of university student startup companies, resulting in 451 valid questionnaires collected.

Due to the insufficient and fragmented information on the number of enterprises in the official announcement on college student entrepreneurship demonstration bases, this study obtained the number of enterprises in each base through online inquiries through the relevant official websites and by making phone calls to the management team responsible for the college student entrepreneurship demonstration bases. In order to ensure the accuracy of the data, I also contacted the leaders of the Human Resources and Social Security Bureau and the Administration for Industry and Commerce, or used online services, and consulted the student employment and entrepreneurship guidance departments at each university and official website information. After comprehensive consideration, this study obtained the number of enterprises that met the research criteria for each base, which was approximately 1100 in Shenzhen, approximately 600 in Shenyang, approximately 840 in Xi'an, approximately 920 in Chongqing, and approximately 1100 in Beijing, totaling 4560 enterprises. These parks represent the majority of universities and follow the principles related to government guidance, social participation, and market operation. They primarily focus on key groups, such as university students, and provide guidance and comprehensive services for graduates, serving as important sources for the aggregation of information on the entrepreneurial elements in universities. Multiple methods, such as telephone inquiries, querying official websites, on-site visits, and document requests, were used to collect and compile this data.

This study used a simplified proportional formula in the quantitative study. Yamane (1967) developed a streamlined formula for estimating the sample size, which is as follows (the formula takes a 95% confidence level and $p = 0.5$ into account [maximum variability]). Simultaneously, considering the sample size required for conducting research using structural equation modeling, as well as the presence of invalid data. This study distributed 605 questionnaires, which was guaranteed to have over 400 valid responses (Kim and Jeong 2018).

$$n = \frac{N}{1 + N(e)^2}$$

This study adopted a stratified proportionate sampling method, based on urban distribution, as the sampling criterion (Zhuang et al. 2020). Among the 4560 companies in the aforementioned cities, 605 entrepreneurs were selected for the questionnaire survey. After excluding invalid and non-responsive questionnaires, a total of 451 valid questionnaires were obtained. The purpose of this survey was to understand respondents' understanding of the research variables, influencing factors, and pathway methods related to business model innovation. The purpose of stratified sampling was to improve the representativeness of the sample by reducing the differences between the units within each stratum and increasing commonality, facilitating the selection of representative samples. The calculation formula for stratified proportionate sampling (Zhuang 2022) is as follows:

$$ni = (Ni/N) \times n$$

### 3.1.2. Control Variables

To mitigate the potential interference of other variables on the research results, it is essential to control influencing factors at both the individual and enterprise level, enhancing the accuracy and generalization of the research findings. Specifically, we controlled for entrepreneur age, entrepreneur education level, and entrepreneur gender at the individual level, and controlled for enterprise age, enterprise scale, and enterprise region at the enterprise level. Differences in the entrepreneur age reflects the variations in experience, ability, thinking, and preferences, all of which may impact how entrepreneurs select new ventures, utilize digital capabilities, and engage in innovation and creation activities (Ferreira et al. 2019). Entrepreneurs' educational backgrounds mirror their work attitudes, values, and management styles (Zhu 2015), potentially influencing their information processing style, knowledge reserve level, work efficiency, and relationship networks, thereby affecting the process of digital opportunity discovery.

Because entrepreneurs of different genders vary in regard to their entrepreneurial motivation, pursuit of goals, job involvement, and social status (Zhu 2015; Wu et al. 2021), we controlled for this variable. At the enterprise level, the enterprise age can impact the strategic orientation of strategic decision-making (Chen 2019). Differences in the scale of new enterprises may influence their access to knowledge and resources, as well as the state, tendency, and ability of enterprises to carry out innovation activities (Lubatkin et al. 2016). Regional differences in policies and economic vitality, where new ventures are located, will affect the importance of enterprises' digital capabilities and technologies (Cai et al. 2019). Therefore, at the enterprise level, we consider the enterprise age, scale, and region, as control variables.

Finally, when measuring the control variables, the entrepreneur's age is determined by their actual age, while the age of the enterprise is determined by its years of establishment. Other control variables are uniformly set using virtual variables. The entrepreneur's educational attainment is determined by their highest educational attainment via a Likert 5-point scale, with 1 denoting 'technical secondary school or less', 2 denoting 'senior high school', 3 denoting 'junior college', 4 denoting 'undergraduate', and 5 denoting 'postgraduate or above.' Entrepreneurs are categorized as male or female, with 1 denoting male entrepreneurs and 2 denoting female entrepreneurs. The enterprise scale is denoted by the number of employees: 1 denotes fewer than 20 employees, 2 denotes 20–50 employees, 3 denotes 50–100 employees, 4 denotes 100–200 employees, and 5 denotes more than 200 employees.

### 3.2. Variable Measurement

Business model innovation: Seven items were adopted to assess business model innovation, starting with the scale used by Luo et al. (2018) and Zott and Amit (2010), which was modified and improved.

Digital capability: Digital capability consists of three dimensions, digital perception ability, digital operation ability, and the ability to collaborate with digital resources. The measurement was based on scales used by Warner and Wäger (2019), Lenka et al. (2017), and Wang et al. (2020), with modifications and improvements. Fifteen items were used in total, with five items in each dimension.

Digital opportunity discovery: Four items were used to measure digital opportunity discovery, using González et al.'s (2017) opportunity discovery measurement scale as a baseline, and modifying and improving it based on the existing literature and the research situation.

Entrepreneur's passion for invention: Five items were employed to assess the entrepreneur's passion for invention, building on and expanding Cardon et al.'s (2013) scale for entrepreneurial passion. The expression of items was optimized considering the research situation.

Enterprise's dynamic capability: Teece's (2007) scale for enterprise perception, absorption, and integration capabilities was used as a reference for measuring the enterprise's

dynamic capability, considering its influence path on business model innovation and the discovery of digital opportunities.

## 4. Results

### *4.1. Empirical Analysis*

4.1.1. Descriptive Statistical Analysis

According to Table 1: Concerning age, more than 30% of the sample falls within the "26 to 35 range. Females constitute 50.33% of the sample, while males make up 49.67%. Regarding educational background, over 40% of the sample have a major related to "*digitization*", and 33.92% have a background in "*business management*". Concerning the years since the company's establishment, over 40% of the sample represents companies established "*4 years*" ago. Concerning company size distribution, the majority of the sample belongs to companies with "*101–200 employees*", accounting for 37.03%. Regarding company locations, the majority of the sample is based in "*Beijing*", with 102 respondents, representing 22.62% of the total. The remaining locations are relatively evenly distributed.

**Table 1.** Frequency analysis results on basic information features in the formal investigation.

| Variable | Option | Frequency | Percentage (%) |
|---|---|---|---|
| Age | 18 to 25 | 90 | 19.96 |
| | 26 to 35 | 163 | 36.14 |
| | 36 to 45 | 104 | 23.06 |
| | 46 to 55 | 59 | 13.08 |
| | 56 and above | 35 | 7.76 |
| Gender | Female | 227 | 50.33 |
| | Male | 224 | 49.67 |
| Education | Business Management | 153 | 33.92 |
| | Other Majors | 101 | 22.39 |
| | Digitalization-related Majors | 197 | 43.68 |
| Years Since Establishment | 2 Years | 38 | 8.43 |
| | 3 Years | 79 | 17.52 |
| | 4 Years | 201 | 44.57 |
| | 5 Years | 117 | 25.94 |
| | ≤1 Year | 16 | 3.55 |
| Company Size | 101~200 people | 167 | 37.03 |
| | 1~20 people | 30 | 6.65 |
| | 200 people and above | 113 | 25.06 |
| | 21~50 people | 41 | 9.09 |
| | 51~100 people | 100 | 22.17 |
| Company Location | Beijing | 102 | 22.62 |
| | Shenyang | 95 | 21.06 |
| | Shenzhen | 102 | 22.62 |
| | Xi'an | 75 | 16.63 |
| | Chongqing | 77 | 17.07 |
| Total | | 451 | 100 |

4.1.2. Reliability Analysis

It is common practice to employ Cronbach's $\alpha$ coefficient in the statistical analysis of empirical research data, as a measure of the survey questionnaire's validity. Typically, if the Cronbach's $\alpha$ value is less than 0.7, it is advisable to re-edit the Likert scale that was created as part of the questionnaire survey. This low value indicates weak internal consistency among the scale's variables.

The Cronbach's $\alpha$ results in Table 2 indicate that the values for each variable in the scale are all above 0.7, demonstrating high internal consistency in the questionnaire. As a result, the dependability of the survey results is excellent.

**Table 2.** Cronbach's reliability analysis.

| Variables | Number of Items | Cronbach α Coefficient |
|---|---|---|
| Digital Perception Ability | 5 | 0.86 |
| Digital Operation Ability | 5 | 0.89 |
| Digital Resource Coordination | 5 | 0.884 |
| Digital Opportunity Discovery | 4 | 0.878 |
| Business Model Innovation | 7 | 0.907 |
| Entrepreneurial Passion for Invention | 5 | 0.873 |
| Perception Ability | 3 | 0.88 |
| Absorption Ability | 3 | 0.877 |
| Integration Ability | 3 | 0.892 |

4.1.3. Validity Analysis

The structural validity of the measurement model was examined in this study through confirmatory factor analysis (CFA). The model's parameters were estimated using the maximum likelihood method. Convergent validity assesses the proximity between the measurement items and their corresponding dimensions. Generally, when the correlation coefficients between the items are high, they are more likely to converge under the corresponding dimension. In statistical terms, both the composite reliability (CR) and the square root of the average variance extracted (AVE) should exceed 0.7 and 0.5, respectively.

The AVE values for each scale are greater than 0.5, and the CR values exceed 0.7, as indicated in Table 3. This demonstrates that the convergent validity of the questionnaire is acceptable.

**Table 3.** Results of AVE and CR indicators for the model.

| Factor | (Average Variance Extracted) | CR (Composite Reliability) |
|---|---|---|
| Digital Perception Ability | 0.554 | 0.861 |
| Digital Operation Ability | 0.620 | 0.891 |
| Digital Resource Coordination | 0.605 | 0.884 |
| Digital Opportunity Discovery | 0.645 | 0.879 |
| Business Model Innovation | 0.583 | 0.907 |
| Entrepreneurial Passion for Invention | 0.579 | 0.873 |
| Perception Ability | 0.710 | 0.880 |
| Absorption Ability | 0.705 | 0.877 |
| Integration Ability | 0.735 | 0.893 |

*4.2. Common Method Bias (CMB) Test*

This study conducted a standard method bias test on the scale, following the single-factor test proposed by Harman. Typically, when the cumulative variance explained by the first component extracted from all the principal components is below 40%, it suggests that the sample data have successfully passed the common method bias test.

The cumulative variation explained by the first component, as shown in Table 4, is 32.296%, or less than 40%. The bias test using the usual method is, therefore, considered to be valid.

*4.3. Correlation Analysis*

In Table 5, Pearson's correlation coefficients were employed to examine the relationships between *Business Model Innovation, Digital Ability, Digital Opportunity Discovery, Entrepreneurial Passion for Innovation,* and *Entrepreneurial Dynamic Capability*. The analysis unveils the following findings: *Business Model Innovation* demonstrates significant correlations with *Digital Ability, Digital Opportunity Discovery, Entrepreneurial Passion for Inovation,*

and *Entrepreneurial Dynamic Capability*, exhibiting correlation coefficients of 0.495, 0.580, 0.613, and 0.553, respectively. These correlation coefficients indicate a positive association between the other variables and business model innovation.

**Table 4.** Explained total variance.

| Component | Initial Eigenvalues | | | Extraction Sums of Squared Loadings | | |
|---|---|---|---|---|---|---|
| | Total | Variance (%) | Cumulative (%) | Total | Variance (%) | Cumulative (%) |
| 1 | 12.918 | 32.296 | 32.296 | 12.918 | 32.296 | 32.296 |
| 2 | 3.182 | 7.954 | 40.250 | 3.182 | 7.954 | 40.250 |
| 3 | 2.358 | 5.895 | 46.145 | 2.358 | 5.895 | 46.145 |
| 4 | 2.183 | 5.456 | 51.601 | 2.183 | 5.456 | 51.601 |
| 5 | 1.922 | 4.806 | 56.406 | 1.922 | 4.806 | 56.406 |
| 6 | 1.784 | 4.460 | 60.867 | 1.784 | 4.460 | 60.867 |
| 7 | 1.700 | 4.251 | 65.118 | 1.700 | 4.251 | 65.118 |
| 8 | 1.293 | 3.232 | 68.350 | 1.293 | 3.232 | 68.350 |
| 9 | 1.015 | 2.538 | 70.888 | 1.015 | 2.538 | 70.888 |
| 10 | 0.588 | 1.470 | 72.358 | | | |

Method: principal component analysis.

**Table 5.** Pearson's correlations.

| | | Digital Ability | Digital Opportunity Discovery | Business Model Innovation | Entrepreneurial Passion for Invention | Entrepreneurial Dynamic Capability |
|---|---|---|---|---|---|---|
| Digital Ability | Correlation *p*-value | 1 | | | | |
| Digital Opportunity Discovery | Correlation *p*-value | 0.481 ** 0.000 | 1 | | | |
| Business Model Innovation | Correlation *p*-value | 0.495 ** 0.000 | 0.580 ** 0.000 | 1 | | |
| Entrepreneurial Passion for Invention | Correlation *p*-value | 0.493 ** 0.000 | 0.644 ** 0.000 | 0.613 ** 0.000 | 1 | |
| Entrepreneurial Dynamic Capability | Correlation *p*-value | 0.432 ** 0.000 | 0.591 ** 0.000 | 0.553 ** 0.000 | 0.685 ** 0.000 | 1 |

** $p < 0.01$.

### 4.4. Structural Equation Model Analysis

4.4.1. Model Fitting Analysis

The study's hypotheses were evaluated using the structural equation modeling (SEM) feature of the AMOS 23.0 software. When performing path analysis, two aspects must be considered: overall model fit indices and the internal structural fit of the model. Fit indices, such as the comparative fit index (CFI), normed fit index (NFI), adjusted Chi-square (CMIN/DF), and root mean square error of approximation (RMSEA), are the primary metrics for assessing the overall model fit. A structural equation model was constructed using AMOS 23.0, based on the model and hypotheses. The fit indices, along with the corresponding standards and results, are presented in the table below.

The fit indices of the structural equation model all meet reasonable requirements, as depicted in Table 6. A satisfactory alignment between the data and the model is observed when the Chi-square to the degrees of freedom ratio is below 3, the RMSEA is less than 0.08, and other fit indices, such as the RMR, RFI, NFI, TLI, and IFI, all exceed 0.8.

**Table 6.** Model fitting analysis.

| Fit | Index | Standard Result | Acceptable or Not |
|---|---|---|---|
| CMIN/DF | <3 | 1.055 | Acceptable |
| NFI | >0.8 | 0.953 | Acceptable |
| RFI | >0.8 | 0.948 | Acceptable |
| IFI | >0.8 | 0.997 | Acceptable |
| TLI | >0.8 | 0.997 | Acceptable |
| RMSEA | <0.8 | 0.011 | Acceptable |

4.4.2. Effect of the Relationship Analysis

The standardized path coefficient from *Digital Ability* to *Digital Opportunity Discovery* is 0.653, as indicated in Table 7, with a *p*-value below 0.05. This suggests that *Digital Ability* has a substantial positive impact on the discovery of *Digital Opportunities*, supporting the hypothesis. Furthermore, the standardized path coefficient from digital aptitude to business model innovation, which is 0.407 and has a *p*-value below 0.05, provides credibility to the hypothesis. The correlation between digital skills and business model innovation is evidently strong. Additionally, the standardized path coefficient from digital opportunity discovery to business model innovation is 0.384 with a *p*-value below 0.05, demonstrating a significantly positive influence of digital opportunity discovery on business model innovation, supporting the hypotheses H1, H2, and H3.

**Table 7.** Path analysis and hypothetical results.

| Path Relationship | | | Standardized Path Coefficients | S.E. | C.R. | *p*-Values |
|---|---|---|---|---|---|---|
| Digital Opportunity Discovery | ← | Digital Ability | 0.653 | 0.19 | 7.366 | 0.000 |
| Business Model Innovation | ← | Digital Ability | 0.407 | 0.172 | 4.513 | 0.000 |
| Business Model Innovation | ← | Digital Opportunity Discovery | 0.384 | 0.065 | 5.25 | 0.000 |

*4.5. Mediation Effect Analysis*

The Bootstrap technique in AMOS 23.0 was employed to investigate the mediation effect of *Digital Opportunity Discovery* between *Digital Ability* and *Business Model Innovation*. The confidence level for the interval was set at 95% (typically set at 90%, 95%, or 99%), and the sample size was specified as 5000 (usually required to be above 1000). The bias-corrected confidence intervals for the upper and lower limits were examined. A mediation effect is deemed present if the bias-corrected confidence interval of the indirect impact does not include zero. Table 8, presented below, summarizes the analysis of the mediation effect.

**Table 8.** Mediation analysis.

| Mediation Path | Effect Type | Value | Lower | Upper | *p* |
|---|---|---|---|---|---|
| Digital Ability → Digital Opportunity Discovery → Business Model Innovation | Direct Effect | 0.407 | 0.059 | 0.855 | 0.024 |
| | Indirect Effect | 0.251 | 0.033 | 0.546 | 0.034 |
| | Total Effect | 0.658 | 0.465 | 0.865 | 0.001 |

The conclusion derived from Table 8 is that none of the 95% confidence intervals for the direct effect, indirect effect, and total effect of the mediation path "*Digital Ability* ≥ *Digital Opportunity Discovery* ≥ *Business Model Innovation*" encompasses zero. This indicates that partial mediation is the form of mediation, and the mediation paths are significant. Therefore, these results support hypothesis H4.

*4.6. Regulatory Effect Analysis*

4.6.1. Regulatory Effect Analysis: Entrepreneur's Passion for Invention

The moderation analysis comprises three models, as illustrated in Table 9. *Digital Ability* serves as the independent variable in Model 1, with *Age, Gender, Educational Background, Years of Establishment, Company Size,* and *Company Location* as the six control variables. *Entrepreneurial Passion for Innovation* is introduced as a moderator variable in Model 2, and the interaction term (the product of the independent variable and the moderator variable) is incorporated in Model 3.

**Table 9.** Moderation analysis results.

|  | Model 1 | Model 2 | Model 3 |
|---|---|---|---|
| Digital Ability | 0.707 ** −11.1 | 0.307 ** −4.98 | 0.408 ** −6.593 |
| Entrepreneurial Passion for Innovation |  | 0.611 ** −13.283 | 0.610 ** −13.751 |
| Digital Ability * Entrepreneurial Passion for Innovation |  |  | 0.256 ** −5.854 |
| Constant | 3.795 ** −12.678 | 3.638 ** −14.342 | 3.457 ** −14.02 |
| Age | −0.107 ** (−2.961) | −0.088 ** (−2.876) | −0.072 * (−2.441) |
| Gender | −0.099 (−1.179) | −0.066 (−0.927) | −0.056 (−0.823) |
| Education | 0.036 −0.64 | 0.091 −1.888 | 0.079 −1.689 |
| Years of Establishment | −0.001 (−0.034) | 0.018 −0.513 | 0.021 −0.629 |
| Company Size | 0.045 −1.215 | 0.013 −0.403 | 0.025 −0.83 |
| Company Location | −0.023 (−0.801) | −0.022 (−0.922) | −0.016 (−0.703) |
| $R^2$ | 0.253 | 0.466 | 0.505 |
| Moderated $R^2$ | 0.241 | 0.457 | 0.495 |
| F Value | $F_{(7,443)} = 21.444, p = 0.000$ | $F_{(8,442)} = 48.249, p = 0.000$ | $F_{(9,441)} = 49.924, p = 0.000$ |
| $\Delta R^2$ | 0.253 | 0.213 | 0.038 |
| $\Delta$F Value | $F_{(7,443)} = 21.444, p = 0.000$ | $F_{(1,442)} = 176.438, p = 0.000$ | $F_{(1,441)} = 34.271, p = 0.000$ |

Dependent variable: Digital Opportunity Discovery; * $p < 0.05$; ** $p < 0.01$ (t-values are shown in parentheses).

Table 10 indicates that the interaction term between *Digital Ability* and *Entrepreneurial Passion for Innovation* is statistically significant (t = 5.854, $p = 0.000 < 0.05$). This implies that when examining the impact of *Digital Ability* on *Digital Opportunity Discovery*, the moderating variable (*Entrepreneur's Passion for Innovation*) exhibits a significant difference in the magnitude of influence at different levels. Subsequent grouped slope analysis reveals that when the moderating variable is at a high level, the regression coefficient between the independent variable and the dependent variable is 0.637, which is significantly higher than the regression coefficient of 0.18, when the moderating variable is at a low level. Therefore, this moderation effect is positive, supporting hypotheses H5, H7, and H8.

**Table 10.** Simple slope analysis.

| Moderator Level | Coefficient | Standard Error | t | p | 95% CI | |
|---|---|---|---|---|---|---|
| Mean | 0.408 | 0.062 | 6.593 | 0 | 0.287 | 0.53 |
| High Level (+1SD) | 0.637 | 0.082 | 7.775 | 0 | 0.476 | 0.797 |
| Low Level (−1SD) | 0.18 | 0.063 | 2.846 | 0.005 | 0.056 | 0.304 |

### 4.6.2. Regulatory Effect Analysis: Dynamic Capability of the Enterprise

Table 11 illustrates that the interaction effect between *Digital Opportunity Discovery* and *Entrepreneurial Dynamic Capability* is statistically significant (t = 2.328, *p* = 0.020 < 0.05). This indicates that when evaluating the impact of *Digital Opportunity Discovery* on *Business Model Innovation*, the moderating variable (*Entrepreneurial Dynamic Capability*) demonstrates a significant difference in its influence at different levels. The specific grouping regression coefficient analysis in the table provides further details.

**Table 11.** Moderation analysis results.

| | Model 1 | Model 2 | Model 3 |
|---|---|---|---|
| Constant | 3.590 ** −15.061 | 3.641 ** −16.043 | 3.638 ** −16.108 |
| Age | 0.02 −0.679 | 0.011 −0.379 | 0.009 −0.313 |
| Gender | 0.03 −0.456 | 0.014 −0.22 | 0.003 −0.047 |
| Education | −0.023 (−0.503) | 0.01 −0.22 | 0.013 −0.288 |
| Years of Establishment | 0.014 −0.432 | 0.014 −0.448 | 0.01 −0.338 |
| Company Size | 0.065 * −2.184 | 0.047 −1.643 | 0.044 −1.565 |
| Company Location | −0.022 (−0.952) | −0.021 (−0.961) | −0.02 (−0.943) |
| Digital Opportunity Discovery | 0.490 ** −14.609 | 0.331 ** −8.375 | 0.398 ** −8.156 |
| Entrepreneurial Dynamic Capability | | 0.347 ** −6.864 | 0.353 ** −7.016 |
| Digital Opportunity Discovery*Entrepreneurial Dynamic Capability | | | 0.084 * −2.328 |
| $R^2$ | 0.347 | 0.41 | 0.417 |
| Moderated $R^2$ | 0.336 | 0.399 | 0.405 |
| F Value | F (7,443) = 33.574, *p* = 0.000 | F (8,442) = 38.324, *p* = 0.000 | F (9,441) = 35.009, *p* = 0.000 |
| $\Delta R^2$ | 0.347 | 0.063 | 0.007 |
| $\Delta$F Value | F (7,443) = 33.574, *p* = 0.000 | F (1,442) = 47.110, *p* = 0.000 | F (1,441) = 5.422, *p* = 0.020 |

Dependent variable: Business Model Innovation; * *p* < 0.05 **; *p* < 0.01 (t-values are shown in parentheses).

Based on the findings in Table 12, there is a noticeable variation in the influence of the moderating variable (*Enterprise Dynamic Capability*) across different levels when examining the impact of digital opportunities on business model innovation. The group slope analysis reveals that when the moderating variable is at a higher level, the regression coefficient between the independent and dependent variables is 0.464, surpassing the regression coefficient of 0.332, observed when the moderating variable is at a lower level. Therefore, this moderating effect is positive and supports hypothesis H6.

**Table 12.** Simple slope analysis.

| Moderator Level | Coefficient | Standard Error | t | *p* | 95% CI | |
|---|---|---|---|---|---|---|
| Mean | 0.398 | 0.049 | 8.156 | 0 | 0.303 | 0.494 |
| High Leve (+1SD) | 0.464 | 0.07 | 6.677 | 0 | 0.328 | 0.601 |
| Low Level (−1SD) | 0.332 | 0.039 | 8.45 | 0 | 0.255 | 0.409 |

To ensure the reliability and validity of the data, we made modifications to the measurement tool based on the results of the factor analysis in the data analysis, aiming to enhance its accuracy and credibility. This process involved eliminating irrelevant components. Additionally, we sought evaluations from domain experts or researchers in related fields to assess the content validity and applicability of the measurement tool.

*4.7. Empirical Research Results*

The research findings further validate the interaction and impact mechanisms among the influencing factors, enhance the theoretical research framework, and the research outcomes have successfully achieved the research objectives. All eight hypotheses in this study have been validated. According to the general interpretation of the structural equation model, the data fits the model well, and the model provides a good level of explanation. This study empirically tests from a strategic and dynamic perspective, showing that *Digital Capability, Digital Opportunity Discovery, Entrepreneurial Passion for Invention*, and *Entrepreneurial Dynamic Capability* directly or indirectly influence business model innovation. *Digital Capability* has a direct impact on *Business Model Innovation*, and *Digital Opportunity Discovery* plays an intermediary role in the relationship between *Digital Capability* and *Business Model Innovation. Entrepreneurial Passion for Invention* moderates the relationship between *Digital Capability* and *Digital Opportunity Discovery*, and *Entrepreneurial Dynamic Capability* moderates the relationship between digital opportunity discovery and business model innovation.

## 5. Discussion

### 5.1. Main Findings

Firms possess a keen ability to discern the opportunities arising from digitization. Enterprise dynamic capability empowers firms to flexibly adjust their strategies and business models, adapting to changes and opportunities in the digital environment, while continuously innovating and enhancing existing business models. The dynamics of opportunities and challenges in the digital environment are ever changing, necessitating firms' continuous learning and adaptation. This ability enables them to continuously optimize the discovery of digital opportunities and the implementation of business model innovation, ensuring the maintenance of a competitive advantage and sustained innovation. Consequently, an enterprise's dynamic capability plays a pivotal role in regulating the relationship between digital opportunity discovery and business model innovation. It facilitates the active discovery of digital opportunities, promotes business model innovation, and ensures the continuous adaptation and response to changes and opportunities in the digital environment.

In conclusion, grounded in dynamic capability theory and innovation theory, this study posits that digital capability significantly and positively impacts business model innovation (research question 1). The exploration of digital opportunities, coupled with the entrepreneurial passion for invention and an enterprise's dynamic capability, emerge as crucial influencing factors in the relationship between digital capability and business model innovation. Digital capability effectively regulates the impact of the entrepreneurial passion for invention on digital opportunity discovery. Subsequently, under the positive influence of an enterprise's dynamic capability, it fosters the mechanism of business model innovation (research question 2). This research model elucidates the boundaries of business model innovation driven by digital capability in university student startups (research question 3). From the perspective of entrepreneurial spirit and the enterprise environment, university student startups grapple with relatively limited resources, a weak digital capability foundation, and poor digital opportunity discovery ability. To achieve the goal of business model innovation through digital capability, leveraging the moderating roles of the entrepreneurial passion for invention and the enterprise's dynamic capability become crucial.

### 5.2. Theoretical Contributions

This study is centered on the digital context and formulates a theoretical model to explore the influence of digital capabilities on business model innovation. The theoretical framework draws on pertinent theories, such as dynamic capability theory and innovation theory. The study also puts forth research hypotheses by referencing the existing literature on digital capabilities, digital opportunity discovery, entrepreneurial passion for invention,

organizational dynamic capabilities, and business model innovation. The objective of this research is to comprehensively unveil the impact mechanism of digital capabilities on business model innovation, along with its boundary conditions. The theoretical innovation of this study is primarily evident in three dimensions:

Firstly, the research extends and enriches both theoretical and empirical investigations into digital capabilities, presenting a novel perspective for studying business model innovation in digital contexts. Current research on the essence and characteristics of digital capabilities lacks depth, particularly in regard to the empirical aspects, resulting in a theoretical lag compared to practical developments (Smith and Johnson 2020). This study delves into the core and nuances of digital capabilities, broadening the understanding on and features of digital capabilities. This expansion contributes to a more comprehensive exploration of digital capabilities and establishes the groundwork for future research. The deconstruction of its essence and features, aids in uncovering the process by which university student startups reconstruct their digital capabilities for survival and development. Furthermore, it offers substantial theoretical backing for the continuous cultivation of new capabilities and models in university student startups, thereby advancing the theoretical research on dynamic capabilities in digital contexts.

Secondly, this study dissects the impact mechanism of digital capabilities on business model innovation. Digital capabilities facilitate timely adjustments to organizational structures for startups, driving the design and innovation of business models aligned with digital development (Smith et al. 2021). Consequently, this paper delves into the connotation and characteristics of digital capabilities, while empirically investigating the relationship between digital capabilities and business model innovation. This not only addresses the imperative to advance the application of dynamic capability theory in digital environments and explore the application of digital capabilities in entrepreneurship (Smith et al. 2021), but also provides a foundation for future in-depth research on digital capabilities and business model innovation in university student startups.

Thirdly, this study elucidates the concept and connotations of digital opportunity discovery, integrating digital elements into the exploration of entrepreneurial opportunities. It enriches the perspective on opportunity discovery research and broadens the application of entrepreneurial opportunity theory in digital contexts. The study is a response to and innovation upon key issues related to digital opportunity discovery. On one hand, the sources, characteristics, and acquisition methods of entrepreneurial opportunities have undergone profound changes, resulting in an increasing number of digital opportunities. This study extensively discusses the connections and differences between digital opportunities and traditional entrepreneurial opportunities, clarifying the conceptual connotations and boundary scope of digital opportunity discovery, thereby laying the foundation for subsequent research. On the other hand, this study introduces digital opportunity discovery into the relationship between digital capabilities and business model innovation, exploring the pathway mechanism through which digital capabilities influence business model innovation.

Digital capabilities enable businesses to better identify and evaluate digital opportunities, proficiently apply and develop digital technologies, generate digital opportunities that align with and lead market demand, and foster business model innovation. This article elucidates the mediating role of digital opportunity discovery and unveils the pathway mechanism through which digital capabilities influence business model innovation. This not only extends the theoretical and empirical research on digital opportunities, but also enhances the innovative empirical research conducted by Smith et al. (2021) on the relationship between digital opportunities and enterprise business models, based on research on startups by university students.

Fourthly, this study delves into the moderating role of entrepreneurial passion in the process of applying digital capabilities to business model innovation in startups by university students, enriching relevant research on entrepreneurial passion in the digital entrepreneurship context. Entrepreneurial passion, as a positive and intense emotion of

entrepreneurs, propels their full engagement, heightens their proactiveness, and amplifies the facilitating effects of digital capabilities on digital opportunity discovery and creation. On one hand, entrepreneurial passion is advantageous for discovering new digital opportunities and reinforces the positive impact of digital capabilities on digital opportunity discovery. On the other hand, passionate entrepreneurs are more likely to create additional digital opportunities. The empirical analysis of the moderating effect also demonstrates that entrepreneurial passion positively moderates the relationship between digital capabilities and digital opportunity discovery. This contributes to a deeper exploration of the mechanisms and boundaries of the influence of digital capabilities.

Fifthly, this study investigates the positive moderating role of dynamic capabilities in digital opportunity discovery and business model innovation. It represents an innovative research direction and has been confirmed by this study, offering a new pathway for research on how startups can drive business model innovation through digital capabilities. This clarification of boundaries establishes a theoretical foundation for research on how digital capabilities drive business model innovation. In the digital age, where opportunities and changes occur rapidly, companies require keen insights and rapid response capabilities to identify and seize digital opportunities. Dynamic capabilities assist companies in promptly discovering new digital opportunities and adjusting their strategies and resource allocation to adapt to market changes. Concurrently, business model innovation is crucial for companies to gain a competitive advantage in the digital age, and dynamic capabilities effectively moderate business model innovation for companies.

### 5.3. Managerial Implications

Digitization is an unavoidable prerequisite for enterprise development and competitiveness. New startups, characterized by small-scale and low-risk resistance, must integrate new digital entrepreneurial elements and explore the digital attributes of traditional entrepreneurial elements to seize opportunities in digital development. In light of this, this study constructs a theoretical framework encompassing digital capabilities, digital opportunity discovery, entrepreneurial inventiveness, enterprise dynamic capabilities, and business model innovation. It provides theoretical guidance for startups to overcome the inertia of traditional entrepreneurial behavior in the digital environment and underscores the significance of digital entrepreneurial activities. Additionally, it offers practical insights for the innovation and development of new startups initiated by university students.

Firstly, startups should prioritize the importance of digital capabilities and proactively cultivate and enhance their own digital proficiency. Startups value digital capabilities to distinguish themselves in market competition, enhance operational efficiency, achieve innovation and transformation, and improve the customer experience. Digital capabilities have become a crucial success factor for enterprises. Furthermore, the research results indicate that digital capabilities effectively promote business model innovation in enterprises. They provide direction for the digital development of enterprises and drive them to innovate their business models. Therefore, startups should consistently learn, cultivate, and apply their digital capabilities, based on their existing characteristics. This approach will encourage startups to actively expand their markets and value networks, discover new digital opportunities, and develop competitive business models.

Furthermore, new startups should also emphasize the pivotal role of digital opportunity discovery in the correlation between digital capabilities and business model innovation. The amalgamation of new and traditional elements is a significant concern that university student startups need to address. In the digital age, the boundaryless nature of digital entrepreneurship activities has generated diverse digital opportunities, demanding robust digital capabilities from new startups. Hence, university student startups should actively nurture their digital capabilities, concentrate on discovering and seizing digital opportunities, and propel the digitization of their products and services to align with market demand.

Additionally, new startups should also recognize the intermediary role of digital opportunity discovery in the connection between digital capabilities and business model innovation. Digital opportunity discovery enhances organizations' and individuals' comprehension of market and customer needs, providing additional channels and platforms for diversified business models. It also compels businesses to innovate their business models by redesigning and integrating their resources and capabilities to adapt to new trends and opportunities in the digital era. In conclusion, digital opportunities play a critical role in digital capabilities and business model innovation, equipping organizations and individuals with more tools and opportunities to better adapt to and leverage the new demands and opportunities of the digital age.

The dynamic capabilities of enterprises play a positive regulatory role in the discovery of digital opportunities and the innovation of business models. This conclusion, affirmed by this study from a new perspective, is manifested in four aspects:

Firstly, through astute market insights: Dynamic capabilities enable companies to promptly identify digital opportunities in the market. Being sensitive to market trends, competitors, and consumer demands allows companies to quickly recognize new digital opportunities and take prompt action.

Secondly, through flexible organizational structures and processes: Dynamic capabilities enable companies to swiftly adjust their organizational structures and processes to adapt to changes in digital opportunities. With flexible organizational structures and processes, companies can better respond to market demand, accelerate decision-making and execution, and seize digital opportunities more effectively.

Thirdly, through innovative thinking and culture: Dynamic capabilities enable companies to foster innovative thinking and culture. By encouraging employees to generate new ideas and solutions, companies can continuously drive business model innovation. Innovative thinking and culture help companies discover new digital opportunities and find innovative business models for business growth.

Fourthly, through robust partnerships: Dynamic capabilities enable companies to establish strong partnerships for the joint development and promotion of digital opportunities. Through collaboration with other companies, startups, and technology providers, companies can share resources and knowledge, accelerate the development of digital opportunities, and innovate business models. In conclusion, college students' new enterprises should prioritize leveraging keen market insights, flexible organizational structures and processes, innovative thinking and culture, as well as strong partnerships to better seize digital opportunities.

Finally, the entrepreneurial passion for invention is one of the key factors driving innovation. Enthusiasm for problem-solving and improving existing products or services can inspire team members' creativity and innovative thinking, fostering continuous progress and development of the company. Through continuous innovation and the introduction of new products or services, companies can offer unique value propositions, attract more customers and market share, meet market demand, and create new market opportunities. By continuously innovating and introducing products or services that meet consumer needs, companies can expand their market size, increase sales and profits, and gain a competitive advantage, while fulfilling market demand.

Additionally, the entrepreneurial passion for invention in new startups led by college students can enhance the company's image and reputation. Through innovation and invention, companies can demonstrate their technical prowess and creativity, establish a professional, reliable, and competitive image, and attract more partners and investors. Therefore, college student startups should prioritize the entrepreneurial passion for invention, concentrate on stimulating team members' creativity and innovative thinking during the innovation process, and consistently propel the company's development and progress.

Theoretical foundations have been laid for the design and operation of the digital framework for the business model of new enterprises, providing actionable strategies. Furthermore, teachers in courses on business management or digitalization can emphasize

the definition, connotation, and operational mechanisms of related variables, such as digital technology, digital opportunity discovery, and business model innovation from the perspectives of innovation theory and the dynamic environment. This will establish a new theoretical framework for the digital promotion of business model innovation, enabling the operational system to recieve advanced guidance. Ultimately, this will help college students in new businesses achieve digital development.

## 6. Conclusions

This paper presents the results of the study based on a literature review and analysis. Firstly, it is argued that in the digital economy environment, business model innovation in university student startups can be achieved through the mediating role of digital opportunity discovery, thereby driving business model innovation through digital capabilities.

Secondly, digital opportunity discovery plays an intermediary role between digital capabilities and business model innovation. Many startups utilize digital technologies to identify new market opportunities, develop and meet market demand (Von Briel et al. 2017), creating new opportunities for innovation, and leveraging the inherent value of digital opportunities to drive business model innovation. Individuals or organizations with high digital capabilities are better able to discover and leverage digital opportunities. They can understand and apply emerging digital technologies to create new business value and address issues within existing business models.

Thirdly, entrepreneurs' passion for invention positively regulates the relationship between digital capability and digital opportunity discovery. This helps entrepreneurs coordinate their goal-oriented cognition and behavior, thereby identifying entrepreneurial opportunities and fostering business innovation.

Additionally, the passion of entrepreneurs also motivates them to actively seek and discover digital opportunities. Their insights into the market and industry enable them to identify potential digital opportunities. They can recognize issues and pain points within existing business models and address them through the application of digital technologies, thereby creating new business value.

Fourthly, the enterprise's dynamic capability positively regulates the relationship between digital opportunity discovery and business model innovation. By identifying and leveraging digital opportunities, dynamic capability can help firms achieve competitive advantages. such as financial and market advantages (Xiao et al. 2019), influencing business models and impacting firms' innovation performance (Luo and Tang 2019; Wu 2015).

This study still has certain limitations and shortcomings, which need to be improved and further explored in subsequent research. Firstly, this study explores and enriches the research on the moderating role of dynamic capabilities in enterprises, clarifying the relationship between dynamic capabilities and digital opportunity discovery and business model innovation through theoretical and empirical research. Business model innovation is a long-term dynamic process, but due to time and resource constraints, as well as the different nature and requirements of dynamic capability adjustment for different types of digital startups, it means that digital capabilities may have different impact processes on different startups. Therefore, future research can further explore the nature and characteristics of dynamic capabilities in different types of enterprises, improve the measurement methods for dynamic capabilities, as well as deeply investigate and comprehensively reveal the dynamic impact and mechanisms of digital capabilities based on different types of digital startups.

**Author Contributions:** Conceptualization, investigation, visualization, writing—original draft: F.L. and K.S. Data curation, formal analysis, methodology, software, validation, writing—review and editing: F.L. Resource supervision: K.S. All authors have read and agreed to the published version of the manuscript.

**Funding:** This research received no specific grant from any funding agency in the public, commercial, or not-for-profit sectors.

**Data Availability Statement:** The datasets generated and/or analyzed during the current study are available, but re-strictions apply to the availability of these data, which were used under license for the current study, and thus cannot be publicly accessed. However, the data can be provided by the authors upon reasonable request and with permission from the third party.

**Acknowledgments:** This article is part of the Doctor of Philosophy in Management program at the College of Interdisciplinary Studies (CBIS) at Rajamangala University of Technology Tawan-ok in Thailand. The researchers would like to thank Khunanan Sukpasjaroen and all the cited experts who have contributed to this research.

**Conflicts of Interest:** The present study has no conflicts of interest with any organization or individual.

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
