# Peer review of "The Impact of the Digital Capability of College Students’ New Enterprises on Business Model Innovation Driven by the Digital Economy: The Mediating Effect of Digital Opportunity Discovery"

_jrfm, doi:10.3390/jrfm17040152_

Round 1

Reviewer 1 Report

Comments and Suggestions for Authors The article describes a comprehensive survey and analysis involving 451 Chinese university student enterprises. The primary objective was to explore the synergistic mechanism between these two factors, dynamic capabilities and business model innovation, to assess their impact and track the development path. The article has a well-defined structure. The individual parts connect to each other and complete a comprehensive picture of the investigated issue. The methodology of the work is described in detail, as well as the results obtained from the research. I appreciate that the authors of the study divided the main findings into theoretical contributions and managerial implications. Such a division is very suitable in view of the investigated issue and the obtained results.   Despite the strengths of this study, there are some shortcomings that need to be addressed. A big negative of this study is the weak focus on the theoretical background. Since this is a very actual issue nowadays, it is a shame, that the authors did not describe other theoretical bases. It will be necessary to complete this chapter and enrich it with knowledge from several sources.

Author Response

We have further elucidated and enriched the theoretical background and development of the variable under study. Thank you very much!

Reviewer 2 Report

Comments and Suggestions for Authors

Well topic for research. 

Comments on the Quality of English Language

I think this paper need to edditing in english.

Author Response

We have once again completed professional English editing and review of the paper. Thank you very much!

Reviewer 3 Report

Comments and Suggestions for Authors

Areas for Improvement:

  1. Literature Review Depth: While the article does present a comprehensive review, it could benefit from a more detailed examination of the nuances within each theoretical concept, especially regarding the evolving understanding of digital capabilities and opportunity discovery in the digital age.
  2. Methodological Transparency: While the empirical methodology is rigorous, greater transparency in the data collection process (e.g., questionnaire design, sampling technique) and analysis (e.g., addressing potential biases, validity, and reliability tests beyond Cronbach's alpha) would strengthen the study's methodological foundation.
  3. Practical Implications: The discussion on managerial implications could be expanded to offer more detailed, actionable strategies for startups and educators in fostering digital capabilities and innovation culture among college students. This would enhance the practical relevance and applicability of the research findings.

Author Response

Dear editor, we have completed the modification according to your suggestion. Thank you very much!
